# Development of a Novel Intra-Operative Score to Record Diseases’ Anatomic Fingerprints (ANAFI Score) for the Prediction of Complete Cytoreduction in Advanced-Stage Ovarian Cancer by Using Machine Learning and Explainable Artificial Intelligence

**DOI:** 10.3390/cancers15030966

**Published:** 2023-02-03

**Authors:** Alexandros Laios, Evangelos Kalampokis, Racheal Johnson, Sarika Munot, Amudha Thangavelu, Richard Hutson, Tim Broadhead, Georgios Theophilou, David Nugent, Diederick De Jong

**Affiliations:** 1Department of Gynaecologic Oncology, St James’s University Hospital, Leeds LS9 7TF, UK; 2Information Systems Lab, Department of Business Administration, University of Macedonia, 54636 Thessaloniki, Greece

**Keywords:** epithelial ovarian cancer, complete cytoreduction, anatomic fingerprints, peritoneal carcinomatosis index, intra-operative mapping, machine learning, explainble artificial intelligence

## Abstract

**Simple Summary:**

The current scoring systems fail to reflect the patient’s real anatomy, as seen by the surgeon upon cytoreduction for advanced-stage epithelial ovarian cancer (EOC). Using artificial intelligence, we developed a novel intra-operative score based on specific weights assigned to the patterns of cancer dissemination. We employed an explainable artificial intelligence (XAI) framework to explain feature effects associated with complete cytoreduction (CC0). The presence of cancer dissemination in specific anatomical sites, including the small bowel mesentery, large bowel serosa, and diaphragmatic peritoneum, could be more predictive of CC0 than the existing scoring tools. Early intra-operative assessment of these areas only accurately predicts CC0 in 9 out of 10 patients and can guide patient selection. The novel score remains predictive of adverse survival outcomes.

**Abstract:**

Background: The Peritoneal Carcinomatosis Index (PCI) and the Intra-operative Mapping for Ovarian Cancer (IMO), to a lesser extent, have been universally validated in advanced-stage epithelial ovarian cancer (EOC) to describe the extent of peritoneal dissemination and are proven to be powerful predictors of the surgical outcome with an added sensitivity of assessment at laparotomy of around 70%. This leaves room for improvement because the two-dimensional anatomic scoring model fails to reflect the patient’s real anatomy, as seen by a surgeon. We hypothesized that tumor dissemination in specific anatomic locations can be more predictive of complete cytoreduction (CC0) and survival than PCI and IMO tools in EOC patients. (2) Methods: We analyzed prospectively data collected from 508 patients with FIGO-stage IIIB-IVB EOC who underwent cytoreductive surgery between January 2014 and December 2019 at a UK tertiary center. We adapted the structured ESGO ovarian cancer report to provide detailed information on the patterns of tumor dissemination (cancer anatomic fingerprints). We employed the extreme gradient boost (XGBoost) to model only the variables referring to the EOC disseminated patterns, to create an intra-operative score and judge the predictive power of the score alone for complete cytoreduction (CC0). Receiver operating characteristic (ROC) curves were then used for performance comparison between the new score and the existing PCI and IMO tools. We applied the Shapley additive explanations (SHAP) framework to support the feature selection of the narrated cancer fingerprints and provide global and local explainability. Survival analysis was performed using Kaplan–Meier curves and Cox regression. (3) Results: An intra-operative disease score was developed based on specific weights assigned to the cancer anatomic fingerprints. The scores range from 0 to 24. The XGBoost predicted CC0 resection (area under curve (AUC) = 0.88 CI = 0.854–0.913) with high accuracy. Organ-specific dissemination on the small bowel mesentery, large bowel serosa, and diaphragmatic peritoneum were the most crucial features globally. When added to the composite model, the novel score slightly enhanced its predictive value (AUC = 0.91, CI = 0.849–0.963). We identified a “turning point”, ≤5, that increased the probability of CC0. Using conventional logistic regression, the new score was superior to the PCI and IMO scores for the prediction of CC0 (AUC = 0.81 vs. 0.73 and 0.67, respectively). In multivariate Cox analysis, a 1-point increase in the new intra-operative score was associated with poorer progression-free (HR: 1.06; 95% CI: 1.03–1.09, *p* < 0.005) and overall survival (HR: 1.04; 95% CI: 1.01–1.07), by 4% and 6%, respectively. (4) Conclusions: The presence of cancer disseminated in specific anatomical sites, including small bowel mesentery, large bowel serosa, and diaphragmatic peritoneum, can be more predictive of CC0 and survival than the entire PCI and IMO scores. Early intra-operative assessment of these areas only may reveal whether CC0 is achievable. In contrast to the PCI and IMO scores, the novel score remains predictive of adverse survival outcomes.

## 1. Introduction

Cancer of the fallopian tube, ovary, or peritoneum (EOC) is the most fatal gynecological malignancy in the western world [1]. Tumor metastasis to the peritoneal cavity is evident in two-thirds of advanced cases at diagnosis (FIGO stage 3–4) [1]. Cytoreductive surgery combined with platinum-based chemotherapy, aiming at total macroscopic clearance (CC0 resection), is the mainstay of treatment. This confers the most favorable outcome irrespective of the surgical timing [2]. In the advanced stages of the disease, extensive surgical procedures are frequently required, including peritoneal stripping, and diaphragmatic, splenic, liver, and gastrointestinal resections [3].

Several assessment tools have been proposed for the extent of peritoneal cancer dissemination [4,5,6,7,8]. For EOC cytoreductive surgery, the Interoperative Mapping of Ovarian Cancer (IMO) score has been widely applied as a simplified tool for precise tumor documentation [5]. Sugarbaker’s peritoneal cancer index (PCI) has been considered the standard, being highly precise and reproducible [9]. It has been universally used for the prognostic assessment or surgical resectability of gastrointestinal carcinomas. The PCI has been long validated in EOC as a significant prognostic factor [10]. Its reproducibility is crucial to evaluate the response of neoadjuvant chemotherapy (NACT) and to decide on the treatment plan [11]. Correlation between PCI and completeness of cytoreduction (CC0) suggests that PCI is a powerful predictor of the surgical outcome. Indeed, a recent study identified a cut-off of ≥24 in relation to achieve CC0 [12]. A PCI ≥ 21 is an independent predictor of severe complications following EOC surgery [13]. In that same study, the added sensitivity of PCI assessment at laparotomy was around 70%. This leaves room for improvement because with the two-dimensional anatomic model for scoring the PCI and IMO fails to reflect the patient’s real anatomy, as seen by surgeons. Moreover, these models lack depth, the anatomic structures in a specific abdominal region may be overlapping, and the imaginary lines may subdivide the regions. An attempt to refine the same anatomical model whilst keeping Sugarbaker’s original concepts led to the development of a modern digital anatomic tool for staging peritoneal surface malignancies [14]. Nevertheless, intra-observer and inter-observer variability decrease the usefulness of PCI [15].

In EOC, tumor localization, rather than the volume of peritoneal dissemination, could be predictive of CC0 [16]. We previously established an advanced-stage EOC database that contains detailed information on the pattern of tumor dissemination (cancer anatomic fingerprints) [17]. In this work, we devised a machine learning-based framework to predict CC0 from composite features, including an exhaustive list of EOC fingerprints for cytoreductive surgery. The study was specifically designed to support the feature selection and weighted importance of the narrated cancer anatomic fingerprints to develop an intra-operative disease score (ANAFI score) and to allow compare it with the PCI and IMO scores for the CC0 prediction. Explainable artificial intelligence (XAI) was applied to explain the predictions by analyzing salient feature interactions. The potential clinical application of the model was finally explored. The primary outcomes were (1) development and performance of a novel ML-based intra-operative disease score to predict CC0 and survival outcomes and (2) development of an XAI methodology to explain the predictions. The secondary outcomes were performance comparison between the new score and the widely used PCI and IMO scores, and evaluation of the impacts on the progression-free survival (PFS) and overall survival (OS) in women with EOC who were operated on at a tertiary UK center.

## 2. Materials and Methods

Our research approach comprised both the development and evaluation of the ANAFI score. In the former phase, we developed a supervised classification machine learning model that predicts complete cytoreduction (CC0) based only on the anatomic fingerprints referring to the EOC dissemination patterns. The predictive importance of each feature determined their weighted importance values in the development of the new score. The latter phase involves the comparison of the ANAFI score with the existing PCI and IMO tools based on (a) receiver operating characteristic (ROC) curves, (b) their performances in predicting complete cytoreduction (CC0), and (c) survival analysis using Kaplan–Meier curves and Cox regression. In both development and evaluation phases, the supervised classification machine learning models were created using the XGBoost algorithm [18], and feature importance were determined using the Shapley additive explanations (SHAP) framework for interpreting predictions based on Shapley values [19].

### 2.1. Selection of Patients, Data Collection, and Study Design

Prospective registered clinical data were initially obtained from 576 consecutive advanced-stage EOC patient records. The cohort included patients who underwent cytoreductive surgery with the intention of CC0 at St James’s University Hospital, Leeds, by a certified gynecological oncology surgeon from January 2014 to December 2019. An internally developed clinical database was integrated with an electronic patient record system [20]. The 2014 International Federation of Gynaecology and Obstetrics (FIGO) classification was used to report staging [7]. Institutional research ethics board approval was obtained through the Leeds Teaching Hospitals Trust (MO20/133163/18.06.20), and informed written consent was obtained. The study was added to the UMIN/CTR Trial Registry (UMIN000049480). Treatment was pre-operatively planned at the weekly central gynecological oncology multidisciplinary team (MDT) meeting prior to patient review.

The hospital setting has been previously described in detail [17]. Women underwent either PDS or three-four cycles of NACT followed by IDS owing to FIGO stage-4 disease; poor performance status (P.S.); or likelihood of being incompletely cytoreduced. Exclusion criteria included patients aged <18 years; those with non-epithelial histology; synchronous primary malignancy; or those undergoing secondary cytoreduction for recurrent disease. Patients with low-grade EOC were included; they were counseled regarding the chemo-resistant nature of the disease, and therefore, the lack of ACT efficacy [21].

### 2.2. Surgical Procedure

All patients underwent the standard institutional therapy for ovarian cancer, namely, primary surgical cytoreduction either in the upfront or the NACT setting. The procedure included as a bare minimum an explorative laparotomy, abdominal hysterectomy with bilateral salpingo-oophorectomy omentectomy, and peritoneal sampling. Additional surgical procedures were performed according to practice recommendations from the British Gynaecological Cancer Society (BGCS) [22]. Colorectal ± upper abdominal surgeons were available ad hoc when maximal effort cytoreductive surgery was required. The 2016 European Society of Gynaecological Oncology (ESGO) for ovarian cancer surgery quality indicators [23] were applied, and an adaptation of the structured ESGO Ovarian Cancer Operative Report was employed to provide detailed information on the patterns of tumor dissemination https://guidelines.esgo.org/, accessed on 15 November 2022. A paradigm shift was initiated in 2016 and 2017 to facilitate more complex multi-visceral surgery performance [17]. At our center, a diagnostic laparoscopy was not routinely offered prior to cytoreductive effort.

Medical record information included date and year of diagnosis, age of the patient, Eastern Co-operative Oncology Group (ECOG), performance status (PS), histology type (low and high), the timing of treatment, the extent of residual disease in diameter, CA125 levels throughout the course of treatment, and survival data (updated in April 2022. Intra-operative variables included tumor anatomic fingerprints, size of the most considerable bulk of the disease, estimated blood loss, and operative time. The PCI and IMO scores were calculated at the beginning of surgery to describe the intra-operative location of the disease [5,9]. The Aletti surgical complexity score (SCS) was assigned to describe the surgical effort [24]. Comprehensive visual assessment of all the areas of the abdomen and pelvis was routinely performed, and no visible residual disease was documented as CC0. Other outcome parameters included overall survival (OS), calculated from the date of diagnosis to the date of death or last follow-up, and progression-free survival (PFS), calculated from the date of diagnosis to the date of confirmed recurrence.

### 2.3. Statistical Analysis

Descriptive statistics are displayed by frequency and percentages for binary and categorical variables and by means and standard deviation (SD) or medians (with lower or upper quartiles for continuous variables). Chi-square and Fisher exact tests were used for categorical and binary variables, as indicated. Logistic regression and receiver operating characteristic (ROC) curves were applied to compare the predictive value of the new score and that of other widely used assessment tools for the completeness of cytoreduction. The Spearman’s correlation r2 was used to examine the association between the new and the existing scores. Survival was analyzed using the Kaplan–Meier and the Mantel–Cox log-rank tests for comparison. All tests were two-sided, and statistical significance was set at *p* < 0.05. The Python programming language was employed for data analysis.

### 2.4. Predictive Model Development and Performance

The extreme gradient boosting (XGBoost) algorithm was employed to predict complete cytoreduction (CC0) based (a) only on the anatomic fingerprints referring to the EOC dissemination patterns and (b) on the ANAFI score, along with the PCI, IMO, and other demographic and operative characteristics. XGBoost is an implementation of a generalized gradient boosting algorithm [25]. Specifically, XGBoost creates new trees that predict the residuals or errors of prior trees and then adds them together to make the final prediction. The novel intra-operative score was based on the specific weights allocated to the cancer anatomic fingerprints.

In both cases, the dataset were split into training and test cohorts (70%: 30% ratio) with repeated random sampling until there was no significant difference (P: 0.20) between the two cohorts with respect to all variables. The training cohort was used to create and fine-tune the predictive model by selecting the set of features that maximized the model’s performance. To this end, five-fold stratified cross-validation (CV) was employed. When performing the CV evaluation, stratified folds were created to ensure the same distribution of negative and positive classes in each fold compared with the entire training dataset. The CV process was iterated 100 times to decrease variance and bias, hence creating and evaluating 500 models in each round. To maximize model performance, we investigated the combined effect of 20 parameters by evaluating a grid of 8000 combinations of values using the Scikit-learn’s GridSearchCV function. The scale_pos_weight hyperparameter of XGBoost tuned the algorithm’s behaviour for imbalanced classification problems. The scale_pos_weight value was used to scale the gradient for the positive class and thus to achieve better performance when making predictions with the positive class. The value of this hyperparameter was set to the imbalance ratio of the response variable in our dataset. To evaluate the model’s effectiveness, we considered multiple metrics that could also capture the balance of the data classes, irrespective of the prediction accuracy. Performance metrics included area under curve (AUC) of the receiver operating characteristic (ROC) curve and the precision–recall curve; and precision, recall, and F1-score for both minority and majority classes.

### 2.5. Model Explainability

To quantify the predictive importance of the variables that were included in the creation of the machine learning models, the Shapley additive explanations (SHAP) framework for interpreting predictions based on Shapley values was employed [19]. Inspired by game theory, it enhances interpretability by computing the importance values for each feature for individual predictions [26]. The method explains the model globally by expressing it as a linear function of features. In other words, it explains how much the presence of a feature contributes to the model’s overall predictions. This method has been already proved particularly important in medical applications [17,27,28] and in policy making [29,30].

To demonstrate the value of the model’s explained CC0 prediction, we developed: (a) SHAP summary plots for the global and local explanation of the results; (b) SHAP dependence plots of the critical risk features for CC0 predictionl (c) SHAP decision plots that explain the CC0 prediction for individual patients.

## 3. Results

### 3.1. ANAFI Score Development

An XGBoost supervised classification machine learning model that predicts complete cytoreduction was created based only on the anatomic fingerprints referring to the EOC dissemination patterns. The complete list of the anatomic fingerprints for the whole dataset broken down by CC0/non-CC0 patients is described in Table A1 in the Appendix A. Moreover, the different XGBoost hyperparameter values that were tested, along with the combinations with the maximum AUC output, are noted in Table A2. The diagnostic accuracy of the developed model to predict CC0 was excellent (AUC = 0.88 CI = 0.854–0.913). Table A3 reports precision, recall, and F1-score for both outcomes (i.e., CC0/non-CC0). Figure A1 depicts the ROC–AUC and precision–recall curves.

The SHAP summary plot is presented as a set of beeswarm plots (Figure 1). The order of the features reflects their importance, i.e., the sum of the SHAP value magnitudes for all the samples. Each point on the summary plot is a Shapley value for a feature and an instance. The position on the y-axis is determined by the feature, and that on the x-axis by a Shapley value. The color represents the value of the feature, from blue (blue) to red (high). An intra-operative disease score was created based on specific weights allocated to cancerous anatomic fingerprints (ANAFI score). In this weighted model, six points were assigned for risk factors with a mean SHAP value of >0.6 (small bowel mesentery disease), five points for risk factors with a mean SHAP value of >0.5 (diaphragmatic disease), four points for risk factors with a mean SHAP value of >0.4 (large bowel serosa disease), and one point for risk factors >0.1 (disease in the appendix and pelvic lymph node involvement). No point was assigned to any risk factors with a mean SHAP value <0.1. Therefore, EOC dissemination on the small bowel mesentery, large bowel serosa, and diaphragmatic peritoneum were the most crucial features globally. The plot indicates the direction of the effects. For example, the presence of disease in the aforementioned three anatomic locations yielded a higher probability of incomplete cytoreduction. In contrast, the presence of disease in the appendix and pelvic lymph nodes translated into a higher probability of achieving CC0. The novel scoring system provides possible scores from 0 to 24 based on the weighted importance of the anatomic fingerprints, which have a median of six in importance.

### 3.2. ANAFI Score Evaluation

#### 3.2.1. Receiver Operator Curves

Spearman’s correlation r2 revealed a closer association between the new score and the IMO score than the PCI (0.74 and O.33, respectively, NS). The new score outperformed the widely used IMO and PCI scores for the CC0 prediction (AUC: 0.81 vs. 0.73 vs. 0.67); see Figure 2.

#### 3.2.2. Predictive Model and Explainability

We thereafter developed a supervised machine learning model to predict complete cytoreduction (CC0) based on the ANAFI score, along with the PCI, IMO, and other demographic and operative characteristics. The list of the features for the whole dataset broken down by training/testing sets and CC0/non-CC0 patients is described in Table A4 in the Appendix B. Moreover, the different XGBoost hyperparameter values that were tested, along with the combination with the maximum AUC output, are noted in Table A6. The diagnostic accuracy of the developed model when predicting CC0 was excellent (AUC = 0.91 CI = 0.864 − 0.943). Table A5 reports precision, recall, and F1-score for both outcomes (i.e., CC0/non-CC0). Figure A2 depicts the ROC–AUC and precision–recall curves.

The importance of features in this model based on SHAP values is depicted in Figure 3. The top-5 list of important features consisted of the ANAFI score, SCS, size of largest bulk of disease, patient age, and pre-surgery CA125.

The SHAP dependence plot reveals the impact of each feature’s value on the prediction problem (Figure 4). It plots the value of a feature on the x-axis and the SHAP feature value on the y-axis by changing a specific feature in the model each time. For the new score, there were three groups of patients with varying predicted surgical outcomes, albeit it appeared that CC0 was most likely to occur for a score below 5 (Figure 4A). There was no inflection point for PCI (Figure 4B). For an IMO score < 5, overall SHAP values were negative, and CC0 was likely to occur (Figure 4C).

For local explainability, the SHAP force plots demonstrate features contributing to pushing from the base value (the average model output over the training dataset) to the model output value. These literal representations of the SHAP values are akin to statistical linear models, where the sum of effects and the intercept equals the prediction. The combination of impacts of all features is the predicted R0 prediction risk. An unusual example referring to an individual case is illustrated in Figure 5; for a given novel intra-operative disease score of 17, provided an SCS of 4, the probability of achieving CC0 was decreased by 2.61-fold. It is slightly improved for an operative time of 215 min but became more likely by 1.62-fold if the patient’s was age 76 years old with the largest bulk of the tumor being 15 cm in size or more. The local explainability example demonstrates the complex interactions between patient characteristics and the surgeon-dependent nature of the cytoreductive surgery, but also that factors globally determining the probability of CC0 may, in extreme cases, be less important for individual patients.

#### 3.2.3. Progression-Free and Overall Survival

The 5-year progression-free survival (PFS) and overall survival (OS) for the total group of 508 patients were 17 months (95% CI:16–18) and 42 months (95% CI = 38–48), respectively. The median PFS was 22 months (95% CI 19–22) for those with CC0 resection and 12 months (95% CI 11–14) for those without CC0 resection (*p* < 0.05) (Figure 6A). The median OS was 57 months for those with CC0 resection (95% CI 54–62) and 30 months (95% CI 27–33) for those without CC0 resection (*p* < 0.05) (Figure 6B).

Progression-free survival was 16 months for patients with ANAFI intra-operative disease scores greater than the median (6). It was 20 months for patients with ANAFI intra-operative scores less than the median (P: $2.72\times 10^{-6}$ ) (Figure 7A). Overall survival was 38 months for patients with ANAFI scores greater than the median (6), and it was 52 months in patients with ANAFI scores greater than the median (P: $3.37\times 10^{-6}$) (Figure 7B). In multivariate Cox analysis, a 1-point increase in the new intra-operative score was associated with poorer PFS (HR: 1.06; 95% CI: 1.03–1.09, *p* < 0.005) and OS (HR: 1.04; 95% CI: 1.01–1.07) by 4% and 6%, respectively. For the PCI and IMO scores, there was a trend toward worsening prognosis, but this was not significant (Table 1 and Figure A3).

## 4. Discussion

Clarity and precision about the anatomical extent of cancer are key to prognostication and cancer-research activities. In this study, we conveyed a picture of the advanced-stage EOC weighted anatomic fingerprints by devising a novel intra-operative disease score (the ANAFI score) that was highly predictive of CC0 (Figure 8). Aside from our intra-operative disease score, CC0 was not solely determined by the technical surgical challenges. Particularly, surgical effort reflected by the SCS, the size of the largest bulk of disease, and the patient’s age were the top-ranking features impacting the likelihood of CC0. The new score is not intended to address the tumor’s biology or patient factors that determine the quality of treatment. Nonetheless, alongside predicting the completeness of surgical cytoreduction, our intra-operative disease score remained the main prognostic feature for survival outcomes.

To date, most of the quantitative intra-operative assessment tools have mainly focused on their predictive value for suboptimal surgery [31], and a valid consensus for predicting CC0 has been slow to establish. Herein, not only did we concentrate on the CC0 forecast, but we also responded to the needs of personalized approaches and embodied the newest technologies, such as ML, to predict surgical and survival outcomes in advanced-stage EOC patients. The ML method has been previously used to predict the resectability of peritoneal carcinomatosis in patients eligible for HIPEC [32]. We adopted a slight modification of the exhaustive ESGO ovarian cancer surgery template to create a more synoptic report, which could serve as an excellent repository to record the attributes of tumor dissemination. Using ML, we then developed a novel assessment tool that can be helpful for EOC peri-operative planning and risk assessments. It appears that the higher the ANAFI score, the more unlikely to achieve CC0. In a cohort with a CC0 rate of 64%, we identified a cut-off ANAFI score ≤5 that makes CC0 achievable. Not surprisingly, the same cut-off applies to the IMO score, likely due to their satisfactory correlation. Conversely, an inflection point ≥10 makes incomplete cytoreduction highly unlikely. That introduces an intermediate group, where ≤5 ANAFI score ≤ 10, which further suggests that unresectability criteria are still based on individual surgical skills and perhaps jurisdictional policies towards cytoreductive surgery [33]. Standardization of surgical practice and identification of centers of excellence for ovarian cancer surgery are required to identify those patients who would benefit from maximal effort surgical cytoreduction to improve their outcomes [34].

Nevertheless, there remains the assumption that tumor distribution can predict surgical outcomes more accurately than tumor size. In contrast to relevant studies devising peri-operative scoring assessment tools, we did not assume that the selected features were equally important when estimating the CC0 prediction output. The rather moderate volumes of the individual anatomic sites allowed us to combine them all into a singular feature score, but also to realistically weight the features’ importance within the specific cohort. (Recall that smaller percentages of the presence of disease in unusual or hazardous locations would not preclude removal to render the patients visibly disease-free.) This supervised feature weighting methodology contributed to the high prediction accuracy of the model. Using the ANAFI assessment tool prior to commencing surgical cytoreduction, CC0 could be successfully predicted in 9 out of 10 advanced-stage EOC women. Even by using logistic regression, our score clearly outperformed the widely used IMO [5] and PCI [9] scores when predicting CC0 in terms of sensitivity and specificity (AUCs 0.81 vs. 0.73 and 0.67, respectively). Since the prime objective remains patient selection for completeness of cytoreduction, we postulate that the new score can be more effective than the more tedious PCI and the more simplistic IMO scores. Nevertheless, humans want explanations. In this respect, we fine-tuned a balance between model accuracy and model interpretability. Being in the era of precision surgery, it was interesting to observe how the global explainability features could apply in individual cases to identify unusual feature contributions. Indeed, the novel score cannot be expected to be perfect, but we envisage that it will demonstrate enduring applicability for wider use in specialized centers for ovarian cancer surgery.

Irrespective of the timing of cytoreduction, it has been shown that completeness of surgical cytoreduction is amongst the strongest predictors of survival in advanced-stage EOC [35,36,37]. In cases of gross residual disease (RD), extensive surgery may be detrimental to survival outcomes [38]. Therefore, our score may be valuable in the early surgical assessment prior to the actual cytoreductive effort in both upfront and delayed surgery settings, even if the chemotherapy response can potentially challenge the discrimination between active cancer and fibrosis. It could be appealing to perform a peri-umbilical laparotomy prior to embarking to surgical cytoreduction for meticulous inspection of all anatomical sites. This would allow the assessment of the three selected regions, since the entire assessment would be of more value based on our findings. During that early assessment of peritoneal dissemination, the scoring threshold should be calculated, which would preclude cytoreductive completeness. This might also be helpful in the reduction of severe operative complications after primary cytoreduction [39].

During surgical exploration, we recorded the extent of carcinomatosis at pre-specified locations (anatomic fingerprints). Undoubtedly, the novel ANAFI score, alongside the IMO and PCI scores, requires a meticulous surgical assessment of all abdominal organs prior to cytoreduction. In contrast to an overall area-dependent assessment, as with IMO and PCI scores, our score is based on the organs affected. At first glance, this may seem complex and time-consuming. However, the main predictors for the macroscopic residual disease were (1) deep invasion of the small bowel mesentery, (2) gross diaphragmatic involvement, and to a lesser extent, (3) large bowel serosa. These three features were more predictive than the entire PCI and IMO scores. What these three regions have in common is that they represent distinct areas of diffuse carcinomatosis, which are apparently difficult to resect. Moreover, widespread mesenteric retractions and small bowel disseminations are unlikely to be detected by preoperative imaging. Several groups have shown that in 80% to 90% of EOC cases of incomplete resection, the residual carcinomatosis was located on the small intestine within the mesentery [16,40]. An Oxford group reported that the incidence of metastatic disease in the peritoneum covering the diaphragm and the liver can be as high as 70% [41]. That same group proposed a systematic classification of diaphragmatic surgery in patients with advanced-stage EOC based on disease spread and surgical complexity [42]. In a previous report from a questionnaire to the members of the Society of Gynecologic Oncologists, bulky diaphragm disease was one of the three most commonly reported sites of disease that resulted in suboptimal cytoreduction [43]. Similar figures were reported for large bowel involvement—up to 70%—suggesting a bowel resection is required to achieve CC0 [44]. In EOC patients, it is difficult to confirm bowel involvement before surgery, likely due to the fact that cancer spreads to the bowel from the external layer and only invades the mucosa at a late stage [45]. This pattern of peritoneal spread for high-grade cancers, referred to as a randomly proximal distribution, has been previously described by Sugarbaker [9]. Frequently assisted by ascites and because of their high capacity for adherence, cancer cells may implant and deeply invade the peritoneal surfaces they initially contact, but they are not likely to migrate great distances from the perforation site. As intracoelomic spread occurs, the wider distribution of the disease makes the complete resection of peritoneal carcinomatosis impossible. Other groups have published similar predictors for achieving CC0 in upfront debulking surgery [16,24]. It has been recently shown that the extent of peritoneal metastases was predictive for an incomplete surgical cytoreduction in the NACT setting [46].

In our study, the anatomical site of the disease was not the only factor for achieving CC0. The surgical effort, represented by SCS [47], was another factor that predicted the completeness of surgical cytoreduction. This is in full agreement with previous studies [40]. Another predictor for complete surgical cytoreduction was the largest bulk of disease. We theorize that size of the largest bulk of disease may represent the aggressiveness or biological behavior of EOC. Although not unchallenged, the size of the initial tumor has been previously reported to be a predictor for the completeness of cytoreduction [24,48]. Patient’s age was amongst the top-5 features predicting CC0 in the multivariate model; in younger patients, this was more likely to be achieved. That said, elderly patients should not be precluded from radical surgery in the advanced stages of the disease [49], but due to the increased likelihood of co-morbidities in those patients, they may be less eligible for aggressive surgical procedures [50]. Our observations are equally supported by the validated Mayo Eligibility Criteria for Surgical Cytoreduction [51]. However, when it came to survival, patients’ age appeared to be less relevant.

In contrast to the PCI and IMO scores, our intra-operative disease score remained predictive of PFS and OS in this advanced-stage EOC cohort. This was not surprising, as the score was also strongly related with CC0. It is common knowledge that prognosis and survival in the advanced stages of EOC are inversely related to RD following cytoreductive surgery [52]. The patient’s prognosis can be improved when the surgeons focus on improving optimal cytoreduction [3]. Even longer OS rates in advanced EOC could be established when no macroscopic residual disease (CC0) was achieved [36].

Amongst the features impacting survival rates, the SCS had an influence on both PFS and OS. This is consistent with previous reports [24]. A recent analysis of SHEER data has shown that women with an upfront surgical cytoreduction had better survival outcomes compared to those who had NACT [53]. To add to the debate, the OS can be compromised by 4.1 months for every course of NACT [54]. Equally, not every patient is eligible for an aggressive upfront surgical approach [55]. In those cases, a treatment strategy consisting of primary chemotherapy was shown to be a reasonable alternative [5,6].

This study benefited from the consecutive patients’ inclusion and the MDT-based treatment strategy performed by the same team of surgeons and oncologists. All data were prospectively collected, and as a consequence, there were no missing values. Nevertheless, the scores were retrospectively assessed. This might have led to some underscoring regarding disease in the small bowel. All women undergoing surgical cytoreduction for advanced ovarian cancer in a catchment area of 3.2 million people were represented in this study. In addition, our study had a substantial number of subjects. The ovarian cancer trajectory is heavily regulated by its multidisciplinary nature; however, despite protocolled selection criteria regarding the timing of attempted cytoreduction, selection bias cannot be excluded. Equally, this should not have had a large influence on the primary outcome, the cytoreducibility. Equally, we did not perform a radiological assessment of the ANAFI score in preoperative CT in analogy to previous work using the PCI score. Radiological PCI appears to not reliably identify patients who are likely to receive CC0 [56]. Interestingly, CT enterography appears superior to routine CT for PCI quantification in these patients [57]. The major limitation of routine CT is the inadequate distention of bowel loops, resulting in serous implants and mesenteric nodules being easily confused with collapsed loops. Despite the obvious advantage of visualization of anatomical regions prior to embarking to cytoreductive surgery (ANAFI score), we are willing to explore the performance of a radiological ANAFI score, and future studies will compare this to pathology reference standards.

Finally, it is worth mentioning that our data were derived from a single institution and may not be generally applicable. To maintain expediency in the ideal world, our novel score requires external validation. Future work will reveal whether the ANAFI score can be predictive of severe post-operative complications.

## 5. Conclusions

The presence of cancer dissemination collectively in specific anatomic sites, including small bowel mesentery, diaphragmatic peritoneum, and the large bowel may be more predictive of CC0 than the entire PCI and IMO scores. Early intra-operative assessment of these areas can guide selection for CC0. An ANAFI score ≥10 is an indication to abort maximal surgical effort for CC0. In contrast to the PCI and IMO scores, the novel score also appears to be predictive of adverse survival outcomes. External validation of our devised score is currently underway.

## Figures and Tables

**Figure 1 cancers-15-00966-f001:**
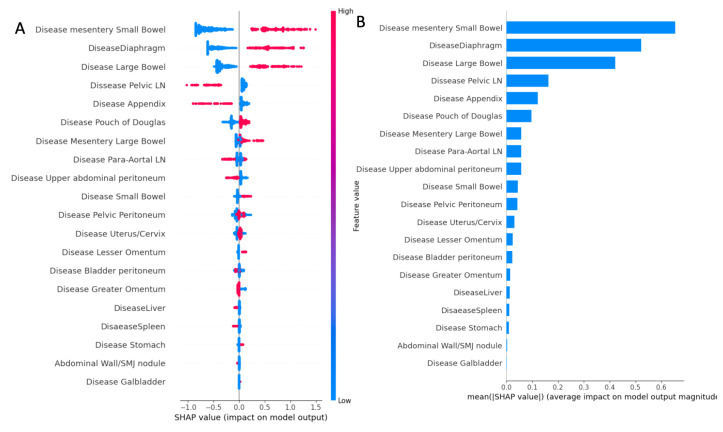
(**A**) Summary plot showing a set of beeswarm plots of feature distribution for global explainability of CC0 prediction based on the anatomic fingerprints of EOC distribution. Dots correspond to the individual EOC patients. (**B**) Feature importance bar plot description of SHAP values for EOC anatomic fingerprints.

**Figure 2 cancers-15-00966-f002:**
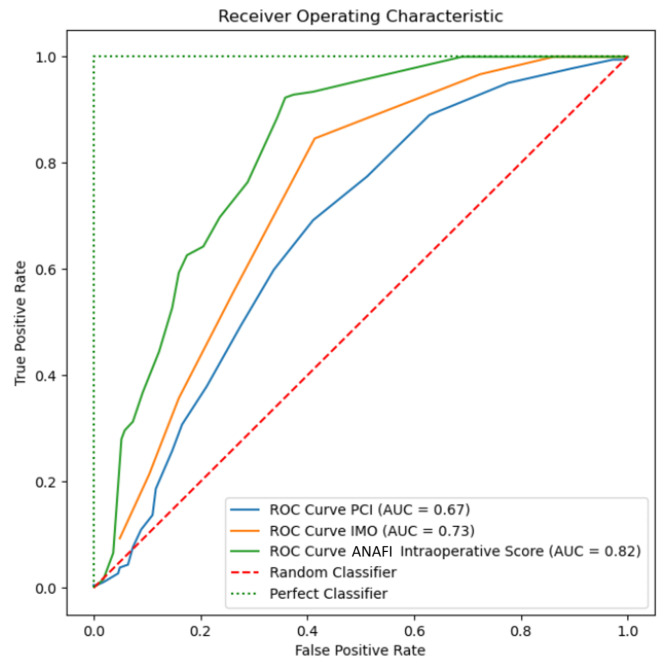
Receiver operator characteristic (ROC) curves for the new score and PCI and IMO scoring systems, showing the estimated probability for complete cytoreduction (CC0). The novel intra-operative disease score based on the weighted EOC anatomic fingerprints can be more discriminant than the existing scores for CC0 prediction.

**Figure 3 cancers-15-00966-f003:**
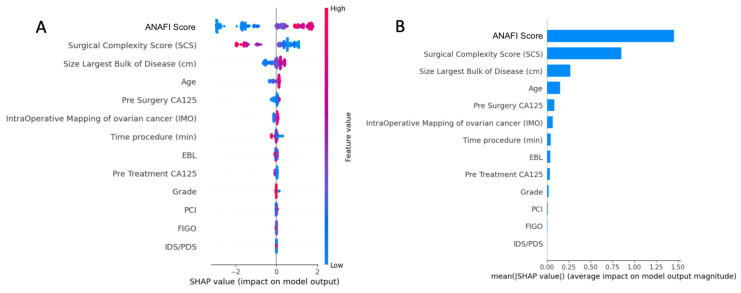
(**A**) Summary plot showing a set of beeswarm plots of feature distribution for global explainability of CC0 prediction, including the novel intra-operative disease score. (**B**) Feature importance bar plot description of SHAP values for the features included in the prediction. Dots correspond to the individual EOC patients. SCS, surgical complexity score; PCI, Peritoneal Carcinomatosis Index; IMO, Intra-operative Mapping for Ovarian Cancer; EBL, estimated blood loss; IDS, interval debulking surgery; PDS, primary debulking surgery.

**Figure 4 cancers-15-00966-f004:**
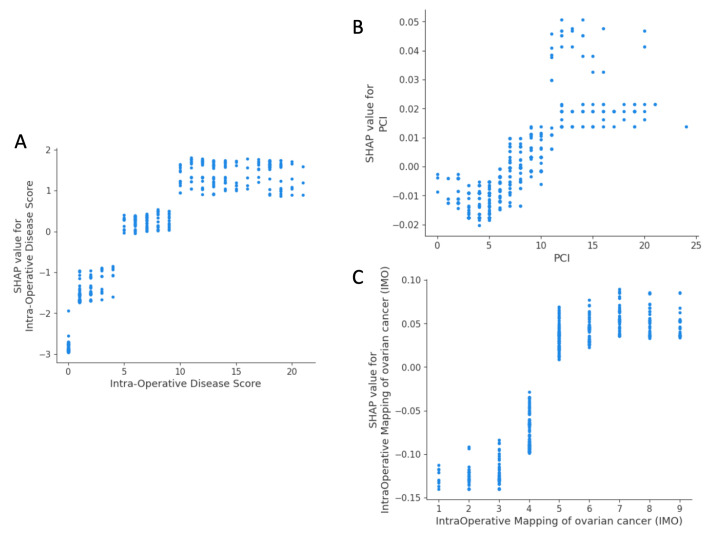
Examples of SHAP value dependence plots for global explainability features describing EOC pattern dissemination. (**A**) Novel intra-operative disease score; (**B**) PCI; (**C**) IMO. PCI, Peritoneal Carcinomatosis Index; IMO, Intra-operative Mapping for Ovarian Cancer.

**Figure 5 cancers-15-00966-f005:**
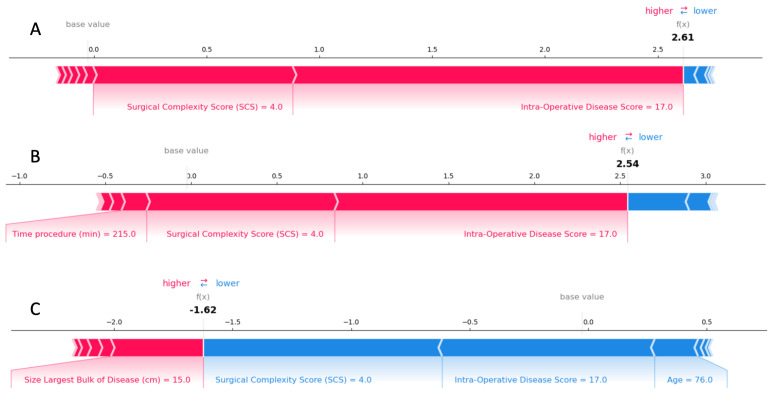
Examples of SHAP force plots illustrating the predicted explained risks for an individual patient. For the R0 resection risk, blue features have values that increased the risk, and red features decreased the risk. The combination of impacts of all features is the predicted R0 prediction risk. In our example, the patient’s predicted risk for CC0 was 2.61. The risk-increasing effect was offset by decreasing effects: consider the ANAFI score of 17, the SCS of 4, the operative time of 215, and the 15 cm size of the largest tumor bulk (**A**–**C**). Subsequently, the odds for CC0 ranged between −1.62 and 2.61 times the expected ones.

**Figure 6 cancers-15-00966-f006:**
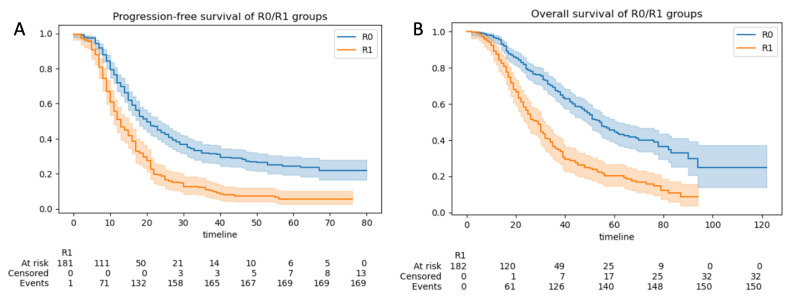
Cohort survival outcomes according to the residual disease (R0 vs. non-R0 (R1)). Kaplan–Meier curves demonstrating (**A**) progression-free survival (PFS) and (**B**) overall survival (OS) analyzed by R0 and non-R0 (R1) groups. The blue line represents the patients with R0 resection, and the orange line those with the non-R0 resections. Confidence intervals (CIs) are illustrated.

**Figure 7 cancers-15-00966-f007:**
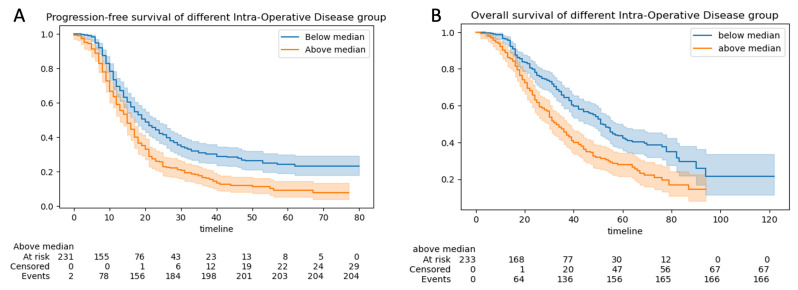
Cohort survival outcomes. Kaplan–Meier curves demonstrating (**A**) progression-free survival (PFS) and (**B**) overall survival (OS) analyzed by >median ANAFI intra-operative disease score and <median ANAFI intra-operative score groups. The blue line represents the patients with <median ANAFI intra-operative disease scores, and the orange line those with >median ANAFI intra-operative disease scores.

**Figure 8 cancers-15-00966-f008:**
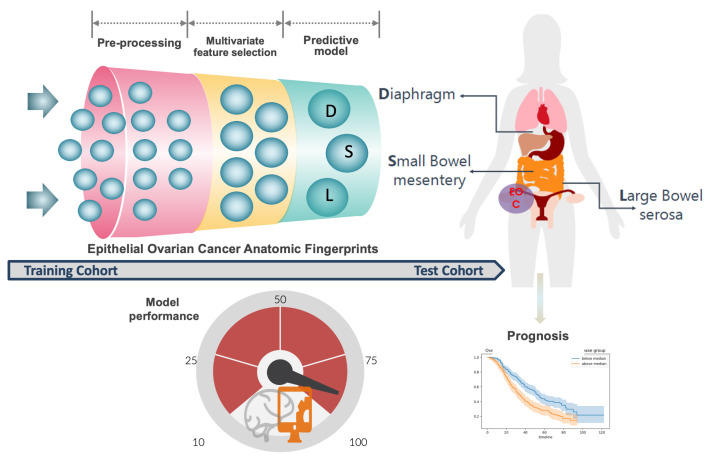
Machine learning-based development of the Leeds Intra-operative disease score. According to our working concept, the presence of EOC dissemination collectively in specific anatomical sites, including small bowel mesentery, large bowel serosa, and diaphragmatic peritoneum, is more predictive of CC0 than the entire PCI and IMO scores. The model’s prognostic value is also demonstrated.

**Table 1 cancers-15-00966-t001:** Multivariate analysis of covariates for progression-free survival (PFS) and overall survival (OS) based on machine learning-based feature selection.

	Multivariate Analysis PFS			Multivariate Analysis OS		
Covariates	HR	*p*	95% CI	HR	*p*	95% CI
Age	1.000	0.53	0.01–0.99	1.000	0.67	0.99–1.01
Grade	1.53	0.06	0.87–0.98	1.32	<0.14	0.92–1.91
PDS/IDS	0.53	<0.005	0.39–0.71	0.61	<0.005	0.48–0.79
Intra Operative Mapping (IMO)	1.04	0.49	0.92–1.18	1.05	0.36	0.94–1.17
Peritoneal Carcinomatosis Index (PCI)	1.03	0.23	0.98–1.08	1.03	0.16	0.99–1.08
Intra-operative Disease score	1.06	<0.005	1.03–1.09	1.04	<0.005	1.01–1.07
Surgical Complexity Score (SCS)	0.88	<0.005	0.83–0.94	0.91	<0.005	0.87–0.96

HR; hazard ratio, CI; confidence interval.

## Data Availability

The data presented in this study are available on request from the corresponding author.

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
