# Peer review of "Development of a Novel Intra-Operative Score to Record Diseases’ Anatomic Fingerprints (ANAFI Score) for the Prediction of Complete Cytoreduction in Advanced-Stage Ovarian Cancer by Using Machine Learning and Explainable Artificial Intelligence"

_cancers, 2023, doi:10.3390/cancers15030966_

Round 1
Reviewer 1 Report
I suggest the authors to provide several figures (such as CT scan, MRI or OP pictures) to illustrate the difference among ANAFI, PCI and IMO tools in EOC patients with complete resection and incomplete resection. It will help the readers easy to catch the points of the manuscript.
Author Response
The ANAFI score is an intraoperative score and the main aim of the study was to introduce the score using ML algorithms and XAI and test it against the PCI and IMO scores. We will be willing to explore the value of a radiological ANAFI score in our follow-on study. We argue the low performance of the PCI score. Herein we highlight the value of a novel intraoperative score based on the weighted importance of intraoperative features early assessed that outperform all composite features in the prediction of complete cytoreduction. A paragraph with references is dedicated in the Discussion section.
Reviewer 2 Report
Dear Authors,
thank you for submitting this manuscript. In the field of gyn onco we still lack optimal tool for patient selection - which is a key moment for successful therapy.
You research is meticulous, scientific sound and convincing. I find it suitable for publication.
Author Response
Many thanks for your positive feedback.